# Shared Brains, Proprioceptiveness, and Critically Approaching the Animal as the Animal in Artworks

Angela Bartram [1,*] and Lee Deigaard [2,*]

1  College of Arts, Humanities and Education, University of Derby, Derby DE22 1GB, UK
2  Independent Scholar, New Orleans, LA 70115, USA
*  Correspondence: a.bartram@derby.ac.uk (A.B.); lee.deigaard@gmail.com (L.D.)

**Abstract:** The animal and being animal is a proposition and position that invites observational and critical debate. Yet, the presence of the non-human animal is usually and normatively confined to representational artworks rather than the animal itself in the gallery or museum, which is, potentially, problematically anthropocentric. Using diverse methods, processes, and materials, and curious to a myriad of opening potentialities, Bartram + Deigaard, in contrast to this problem, explore working as humans from an animal-centric perspective through artistic research. They bring sensitivities to their handling of the animal, as both artistic subject and collaborator, to observe and engage with empathy and openness to animal insight and revelation and behaviour. Their works in performance, video, drawing, and printmaking foreground animal proximity and behaviour, inter-species proprioception, reciprocal caretaking, synchronised respiration, and companionate movement. This article explores the socialised and familiar in close observation, directly and indirectly, in their individual yet companion practices, illuminating the benefits of a radically enlarged sentiocentrism. It reflects on the allowing and embracing of other species within their artworks, and of being mindful and sensible with balancing sympathies and empathies as humans within an often unbalanced system of agency. Specifically, it gleans patterns and insights from their exhibition at Tippetts and Eccles Galleries at Utah State University in 2021, where they invited a canine collaborator into their thinking through praxis and the interventions and residual outcomes this created. This essay discusses two individual video artworks from each artist, which document their invitations to non-human animals into the gallery or museum, and two durational artworks curated within this exhibition.

**Keywords:** proprioceptive; artists; interspecies; collaboration

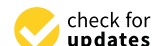



## 1. Introduction

The presence of the non-human animal is usually and normatively confined to representational artworks rather than the animal itself in the gallery or museum. A problematic anthropocentrism, therefore, exists within exhibitions containing images and the bodies of animals. Jacques Derrida, writing about his cat, observes that "as with every bottomless gaze, as with the eyes of the other, the gaze called 'animal' offers to my sight the abyssal limit of the human: the inhuman or the ahuman, the ends of man", and it is this that creates interest in the subject (Derrida 2002, p. 389). As such, there has long been a craving to see animals up close, to experience and observe the other without threat or danger, such as in the taxidermy of indigenous and exotic species shown in glass display cases. This type of practice makes safe the animal, the unknown, in its offer to human observation. Despite the pleasure that many artists take in blurring the boundaries between human and non-human worlds and bodies, in ways that may expose a viewer to questions of the lines that separate species, they often retreat to comfortable ways of making and showing art. A reliance on imagery rather than live representation within exhibitions is welcome, therefore, and artists have responded accordingly.

Artists have historically, and in some cases contemporaneously, reduced the living non-human animal to representation alone. Image makers, as these types of artists employing

animal subject matter are primarily, include the animal only as a codified representational form which becomes inscribed into the artwork as a body alone. What animals can contribute with their minds beyond being material and 'stuff' is generally ignored, their cognitive abilities made irrelevant, as humans exercise coercive control and a seemingly organic artistic dominance. Non-human animals, in this context, function much as with clay, plaster, and other inert making-materials, as their living sensibility is ignored. This is assumed to be desirable and permissible because of pernicious assumptions about difference: one entity is object and image-source, the other is subject and image-maker, resulting in an inevitable, downward spiral of essentialising and reduction in addition to the forfeiture of valuable collaborative input.

The artwork of Bartram and Deigaard regards non-human animals as equals in relation to creative engagement and input in contrast and (perhaps) in challenge to this dilemma. This can be a big deal for curators and organisations, as it takes much negotiation of the normative ways of existing as artists within the gallery and museum system, but, as this essay explores, for them it is worthwhile. For the challenge can create positive, productive, and thoughtful exchange within the exhibition. In 2017 the artists independently recognised a significant number of shared methods and ethical parameters in this respect: they both work collaboratively with non-human animals; they both maintain a sense of being together with non-human animals while conducting research; they both routinely collaborate with non-human animals within mutual, beneficial co-economies of cognition and proprioception (proprioception, and being proprioceptive relates to the sensing and perception of other bodies actions and location in this essay). Their formal collaboration as a creative duo, Bartram + Deigaard, inaugurated in November 2021 when they converged for a joint exhibition in the Tippetts and Eccles Galleries in Logan, Utah.

Lee Deigaard is a multidisciplinary artist based in New Orleans and rural Georgia, USA, with her companion dog, Elvira. She collaborated for 23 years with her horse, Blue, and their relationship was mutually and entirely apart from equestrian activities. She conducts long term observational and artistic research on wild animals across the United States, especially in the south-eastern United States. Angela Bartram is a researcher and artist based in Nottingham, UK, living with four companion cats and a horse. Bartram has explored empathy and the politics of (in)equality with companion animals in artistic research since 2003 when she began collaborating with her dog, Woofer. Bartram and Deigaard have deep and significant experience of animals through long-term relationships with companion species, across many different living situations and shared creative endeavours. Individually and collectively, they follow an antithetical position to Cartesian dualism and Rousseau, who asserted in *Discourse on the Origin of Inequality*, that humans bear a moral responsibility to impose no cruelty on non-human animals exactly due to their inferior status among sentient beings (Rousseau [1755] 1992).

Artistically, and theoretically, and in opposition to the now acknowledged (for most) misrecognition of non-human animals, they creatively research with non-human animals, rather than on them. Their shared ethos is equality centred and mutually balanced, whereby no (human) animal is accorded more importance and agency than the other. This is an inclusive shared approach to artistic practice that is highly sensitive to the other within both making and exhibition contexts; and this drives their individual and their collaborative activities. They aim to know non-human animal companions through respectful and improvisatory engagements based in mutual affinity and empathy, and through acknowledging animal behaviours within their own ways of being and practising as artists. The shared goal of Bartram and Deigaard is to understand and learn, to convey and enable these experiences and animal-shared insights to others through exhibited artistic research.

What follows is a critical unfolding and re-mapping of the concepts, ideas, and intentions for their first collaborative exhibition, *Draw | Breath | Animal*, at Tippetts and Eccles Galleries in Logan, Utah where their critical thinking and connectivity converged in further incubation and coming into being within a shared exhibition space (Figure 1). Within this account is an acknowledgment and contextualisation of collaboration through shared think-



ing, how this is constructed at a distance from their respective countries, and the dynamic necessity for this exhibition to involve the non-human in the form of Deigaard's dog Elvira, in order to bind and connect the project through her investigating physical presence in the gallery and reciprocated empathetic engagement with the two human artists. Following Donna Haraway's statement in *The Companion Species Manifesto: Dogs, People, and Significant Otherness* that, "dogs are not surrogates for theory: they are not here just to think with", her inclusion recognises the potential of the animal to be more-than-the-sum of their body in creativity (Haraway 2003, p. 5).

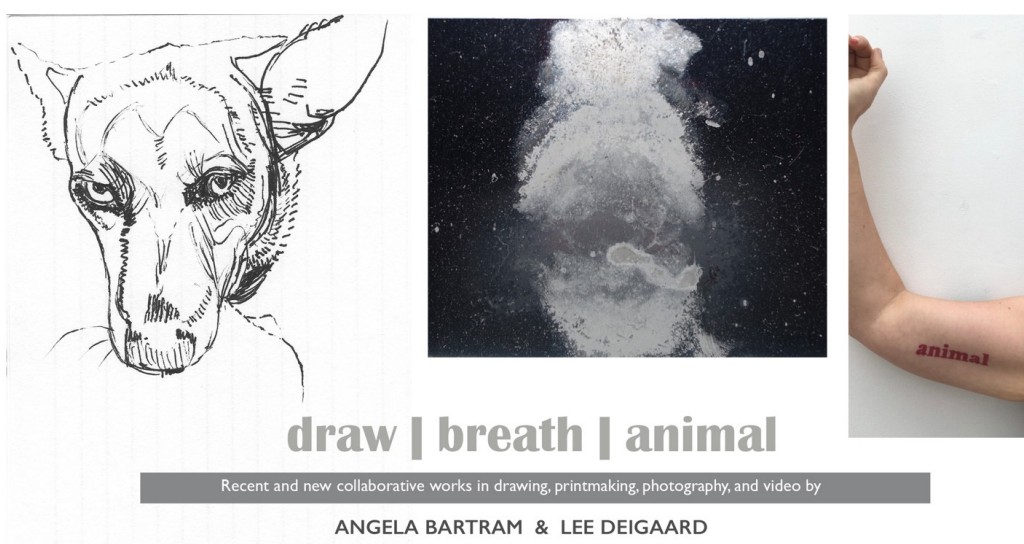

**Figure 1.** *Draw | Breath | Animal*, exhibition poster, Tippetts and Eccles Galleries, 2021. Photo by the artists.

The artwork examined in this paper turns anthropocentric viewing orthodoxies on their head, welcoming and emphasising constructive, mutually supportive inter-species differences within the operational framework of artistic collaboration. Empathy, and being equal and connected as sensing and sensitive (to the other) bodies, irrespective of species derivation, is significant with their collaboration being forged on individual and collective assertions of allowing all to have creative input and agency. Dogs are extraordinary empaths, and it is this ability that allows them to bond, sense, and synchronise so favourable with humans, allowing for their use as human guarding, medical, and assistance animals. The animal behaviourists Charlotte Duranton and Florence Gaunet explain dogs synchronicity as adaption to human life through their ability for empathetic attachment in *Canis Sensitivus: Affiliation and Dogs' Sensitivity to Others' Behavior as the Basis for Synchronization with Humans?*, as being a product of "the special relationship between domestic dogs and their human caregivers . . . [e.g.,] dogs sleep when humans sleep and are active when they are active . . . [leading to their empathy and] sensitivity to others behaviours and the ability to adjust their own behaviour accordingly" (Duranton and Gaunet 2015, p. 514). The artists invite the dog to explore how to be empathetic creatively in the context of an exhibition—Elvira is not coerced, objectified or subjectified, or reduced to imagistic or bodily material, but is welcomed into a collective and reflexive interspecies collaboration of free agency and cognition. The creative outcomes and effects detailed here will be discussed in relation to their responsive intentions for equality through assertive collaboration with other art bodies, behaviours, identities, materials, and processes.

## 2. Articulating the Creative Animal

The conventional perception of the non-human animal imposes a hierarchy of value in which the human occupies a permanently privileged position and the non-human, even when admired, does not. Non-human animals are therefore marked as other within a binary

of them and us, the unprivileged and the privileged, within a sliding scale of intellectual capacity and implication of savagery. Within the context of collaborative artistic research, a very different approach is possible: empathetic regard tied to parity, ethical care, and duty.

The pet and companion, as those brought within close allegiance to humans, inhabits a grey zone within the human to non-human animal lineage. The pet is defined here as the non-human animal that is brought into close and familial relations within domestic life, but where human dominance might be in effect; following Haraway, a companion is defined an extension to the relationship of pet and human, but where equality is necessary—a situation where there has "to be at least two to make one", which sees both human and non-human given equal companionable status (Haraway 2003, p. 12). Living conditions, often within the construct of the human domestic 'home' (cats and dogs, for example) or as an integral component of the self-determined and prescribed construction of one's life (such as a horse), sees the pet granted, as a particular form of human largesse, with close affiliation to the human with which they are attached. Humans invite them to share whilst bestowing care, yet in most circumstances they decide the form, expression, and its duration, which is subject to their rules. Difference is evident, recognised, and enforced through the meanings, gestures, and behaviours of dominance within interspecies companionship. The domestic comes closest to a diminution of hierarchies, as the home is a site of intimacy and familiarity within co-living, but the human conditions this to their will.

The positions of 'other' and 'animal' are significant to artists (it is acknowledged that these, particularly when wielded together, are problematic terms, but they are necessary in this context for their bearing on levels of equality and agency. Any use of the terms henceforth is made with a declaration and sense of their problematic usage). As those who often step outside normalised human conventions, artists adopt a heterogeneous position to better allow creative observation and reflection upon the conditions of being human, permitting them more unfiltered and engaged responses. This stepping out, or aside, even if momentary, self-determines a position with, and of, 'other' and 'animal' by the hierarchical constructs with which they are defined.

Hal Foster wrote in *The Return of the Real* that the artist must take the position of an outsider to be able to observe the world from a position of the ethnographer; they have to partially forfeit their humanity in order to assert and accomplish true observation and critique, so that it is in the "name [of the cultural other] the committed artist most often struggles" (Foster 1996, p. 173). Deleuze and Guattari explored this notion of the outsider through rhizomatic and multi-entry point experience, locating it with any taking on of a becoming animal as a unique process (Deleuze and Guattari 1987, p. 7). Operating within a rhizomatic, material, and animal sensing practice unites around a struggle to interrogate and subvert orthodox hierarchies, not only intellectually, but also physically and in real time within a cultural setting. Being 'animal' in self-determined roles as artists, is to embrace the wild and feral, in subtle antagonism to a human-centric world and a semi-negation of its underlying assumptions. The artist, human and creative, seeking to collaborate and grow through proximity to the non-human animal, is therefore like the pet in this respect, through a temporary inhabiting of closeness and affiliation outside of a defined structure. A duality is initiated and present, whereby the artist is both human-animal and human observer, both within and beyond human conventions. Such a situation, therefore, allows for the potential recognition of the non-human animal with the semi-displaced self. If developed, this recognition can incite a fuller engagement with shared interspecies creative agency and the subsequent insights it elicits. It opens the potential for rhizomic, multi-layered engagement with, and offering to, an audience who through the capacity offered by the creative agitator, might consequently feel differently about the subject.

## 3. Shared Agency

Non-human animals have a lot to offer. Humans, who engage with them daily, know this to be true. In the case of pets, these interactions are purposeful and meaningful; the animal is invited and wanted in daily life. Pets are, through sharing in domesticity and

life, the nearly humans with whom we choose to associate, and who, by being given the rights of closeness to humans, are those we acknowledge as of interest. Creatively, it has been stressed that artists and non-human animals have allegiance, similarity, and synergy by their shared adjacency to humanity. Is there value in acting on an acknowledgement of these shared states?

When offered collaborative inter-species artistic agency, the non-human animal can gain capacity through their awareness of and their potential for creative and cognitive input. This relies on the human artists' willingness themselves to be animal, to approach the non-human with equality in position, thinking, and behaviour, and in being comfortable with shared acknowledgment and recognition. Independently and as a grounding principle, this way of being is central to the creative activities of Angela Bartram and Lee Deigaard. If humans can stop determining the unfolding of events or holding to objectives and performance and performative expectations, one of many key lessons that animals teach us is how to abide within moments and the sensory world more fully. When doing this, you often find them waiting for you in a liminal, emotional, and intuitive space which is where creativity arises and artmaking happens—through aliveness to the world, improvisation, and the unexpected. Central to the methodology of both Bartram and Deigaard is the experiment (with unknown outcome); central also is the experience of the animal collaborator who heightens and deepens the work as participant, audience, protagonist, enactor, teacher, and enabler. This is enhanced by seeking out and inviting open-ended and voluntary ways for non-humans to participate. In such situations, it becomes possible to ask who, in fact, is the partner and who is the creator? Who invites and who participates? Is there a separation or a coalition that is demonstrated in an enabling of the non-human as artistic? Can equality ever be fully realised, even if this is a prioritised intention?

In order for the realisation of equality in an interspecies creative process, there needs to be an acceptance of what animals (human and non-human) will bring. Participation must be freely given by both, and not rendered under duress, implicit or overt, by the non-human at the will of the human. This freedom invites a sense of the unpredictable and the feral, and a relinquishing of human control, but it is necessary if free and untethered involvement is to be allowed. So, how might this happen? To do this, the human artist must accept the animal, and to be acting as animal in turn in their own involvement. This does not mean a refusal of the human, or an absolute shift in stance towards the animal; on the contrary, it is a declaration of the human as animal without the ego and trappings of normative and conditioned human behaviour. An opportunity comes forth from such practice, in which the invited non-human is an unfettered and fully capable participant (Figure 2).

Bartram and Deigaard recognise the feral within themselves and especially in their artistic engagement with those more normally described as 'other.' They individually and collaboratively look closely at what is different to normal constructs within human ways of being, to take on the eyes of the non-human animal (receiving their gazes, considering how they see) in order to represent a different way of looking (and being looked at). Invested in the animal as a way of working and the animal as a collaborative construct, they are committed to exploring what it means to creatively, practically, ethically, and cooperatively work with non-humans. This is specific to pets within this exhibition in the exploration of the benefits that familiarity and closeness of species can bring. They also converge in their collaboration as animals distinct in behaviour and personality from one another, whose individuality informs a duality in which they make a commitment to presence and to being with. The artworks in their exhibition *Draw|Breath|Animal* are accounts of that commitment.

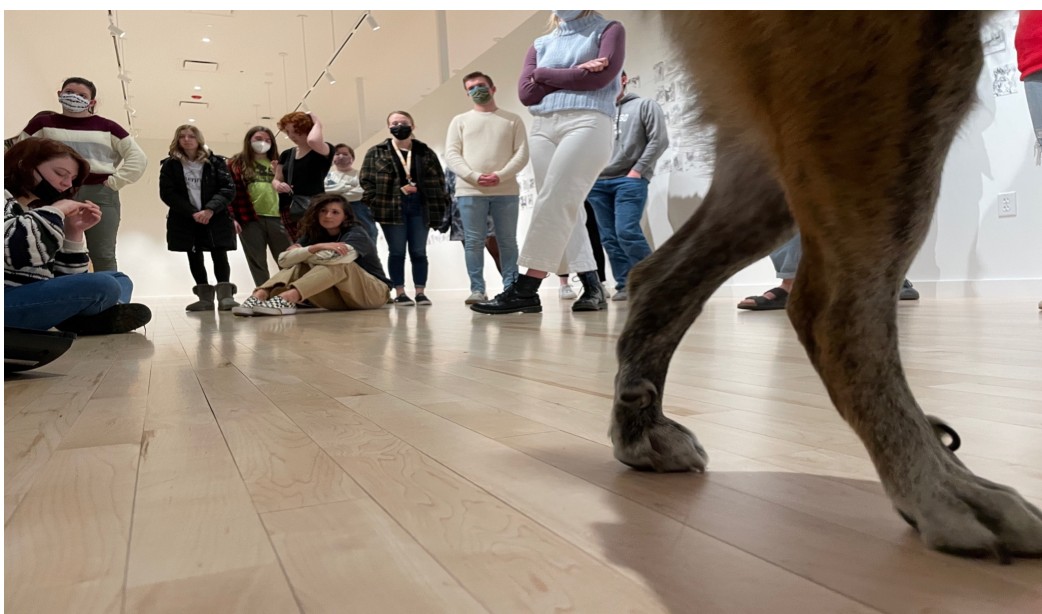

**Figure 2.** Artists talk with Elvira, in the *Draw|Breath|Animal* exhibition, Tippetts and Eccles Galleries, 2021. Photo by the artists.

## 4. Artistic Animal Initiatives and Patterns of Creative Commitment

The opportunity for exhibition and conveying a sense of interspecies sharing to a different community and audience can provide a testing ground for reception, significance, and interest. An exhibition provides an effective vehicle for the artists to stage ideas and methods in a particular way, in their way, in order to encourage engagement and reflection on the subject in communication with, and to, the audience. How and what one stages within the context of an exhibition is of importance; selection is crucial, and artworks should be curated and installed in response to each other as well as the overall thematic scope. This is particularly important when the exhibition is a collaborative initiative; the artworks should harmoniously unfold and clearly tell the reflective engagements therein. Curatorial rules, especially those set out by the artists themselves, as in the case of *Draw|Breath|Animal*, create an effective challenge that informs the reading of intent and subject.

Balancing the needs and requirements of two artists and their artworks can be problematic. Egos can take hold, which in humans can be disruptive or confrontational in particularising creative pursuit and delivery. Often, negotiations concerning space and content seem tricky, out of dialogue and harmony, and unbalanced. Successful collaboration brings respective needs into a hoped-for harmonious dialogue, a relationship concerning overlaps and proximities, even when the displayed artworks have been made individually. *Draw|Breath|Animal* required a collaborative contract, an informal mission statement between the active (human) agents themselves. What to show, and how? What best reflects process and duration and the various intimacies and allegiances involved? Re-telling of artistic interspecies topographies, of time, engagement, and duration requires careful curatorial mapping and taking care of all components. In *Ways of Curating*, Hans Ulrich Obrist reminds us of the "Latin etymological root [of curating] curare: to take care of" ([Obrist 2014](), pp. 24–25). The various synchronicities and analogous content, the plotting of intentions and enmeshing, need to be managed and cared for with sensitivity to meaning and artistic purpose.

The days and nights of the artists, with one located in the UK and the other in the USA, were shifted out of synchrony. Distance is a logistical barrier; it alters means of connecting and shifts simultaneous understandings. There are significant zones of not knowing, of unknowing. The developing and necessary collaboration, therefore, was misaligned in and over time and through distance. Over long distances, communication develops nodes

along overlapping wavelengths, and this opportunity was held within the purpose of the exhibition as a means of intent and intrinsic to the selection of artwork to be included. The artists looked at previous artworks that collectively and intentionally bound them and had never been seen together in dialogue and contiguous space.

The reviewing of video footage after the fact (such as ongoingly with *Be Your Dog*, Bartram, and *Gus and Deuce Go Elsewhere*, Deigaard [Figure 3]) not only teaches and informs us of interactions and animal communications (ears swivelling, body language) that can outpace human perception in real time, but also gives rise to layered and significantly symbiotic artworks and related iterations across disciplines. Bodily engagement and mutual sensory immersion yield insights, reflective and documentary, of acts of companionship shared in (otherwise and normatively human-centric) artistic space. Bartram's *Be Your Dog*, a documentary video of a dog and human artistic pack formation within Karst, a gallery in Plymouth UK, made collaboratively in 2016, explicitly invokes the engagement of bodies in concert and response, and of the human following the animal's lead towards heightened empathetic access. Deigaard's *Gus and Deuce Go Elsewhere* invites the curiosities and disproportionate considerations of horses' free entry into a museum, emphasising their crucial autonomy in choosing to enter or not, and their horse-directed explorations of its contents towards their own enjoyment.

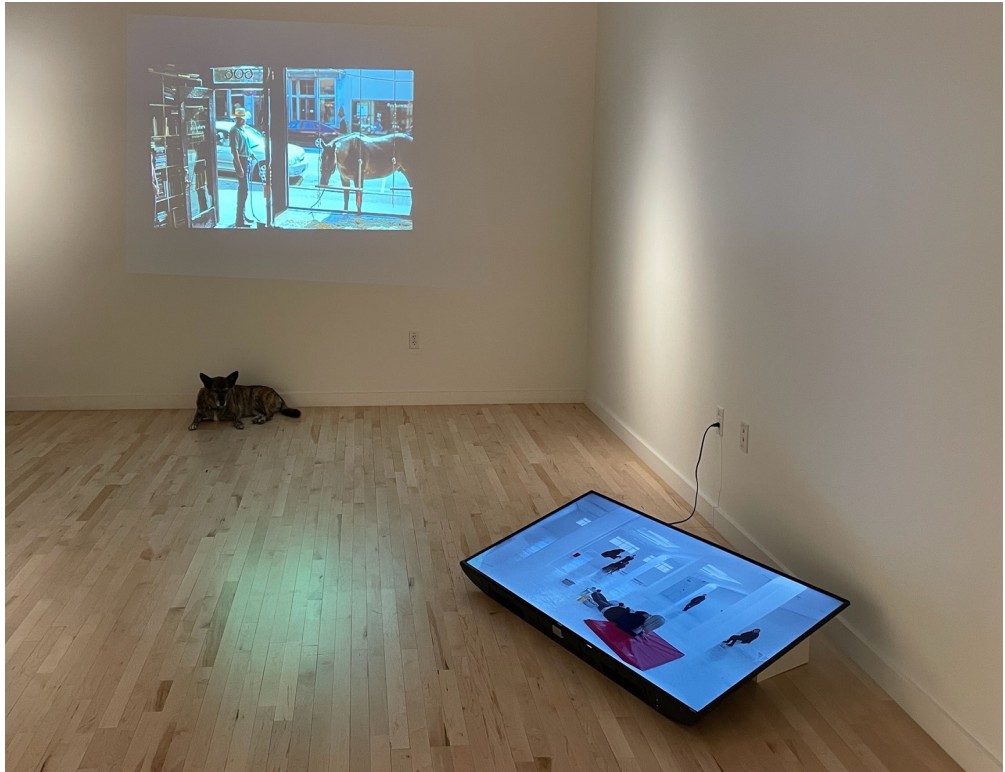

**Figure 3.** *Be Your Dog* (floor monitor) and *Gus and Deuce Go Elsewhere* (wall projection), with Elvira, in the *Draw|Breath|Animal* exhibition, Tippetts and Eccles Galleries, 2021. Photo by the artists.

Both projects sought to enhance the non-human within spaces given to culture and its (implied) human consumption, and to question anthropocentrism and the right of agency within the creative act and its reception. This is in stark and assertive opposition to a long held Cartesian dualism considering animals responsive as bodies alone, and not as minds. True collaborations have a mutualism and a sympathy, a buttressing in the partnering and combining of strengths and efforts, as evident in these artworks. They embed and assertively engage with differences in order to heighten possibilities and explorations through the acknowledgement of, and working through, of mutually conditioned acts of empathy.

## 5. Defining Creative Histories That Bind

The artists share an empathetic connection, both as humans working creatively together through collaboration, and by connecting with the sensibilities of non-human animals. Bartram's and Deigaard's individual artistic research enquiries are both devotional practices pursued with other species, within undefined outcomes, celebrating and marking the ephemeral. Devotional here concerns a service to, and with, the non-human animal, but also is about 'keeping in mind' or 'holding in your heart' their difference.

There is curious and predetermined play within Bartram's *Be Your Dog* that the artist intends as a positional statement affirming equity, empathy, and creative agency within interspecies dialogues (Figure 4). The public performance, as seen in video footage on the floor-mounted monitor within the Logan exhibition, took place within Karst, the host gallery in Plymouth (UK) in November 2016. Seven pairs of dogs and their humans were selected from an open call to create an interspecies artistic pack. Participants were selected based on the human's descriptions and understandings of empathy within their domestic relationships with their dogs. The invitation to participate did not concern human-centric privilege or gratification, and selection was based on the person's willingness not to be dominant in this context but in service to their dog's experience. The participants did not know each other; they did not know the artist. The selected dogs represented many breeds and sizes and were of mixed gender, as were the humans. All brought their own sensibilities. Two consecutive weekend workshops took place before the eventual public performance.

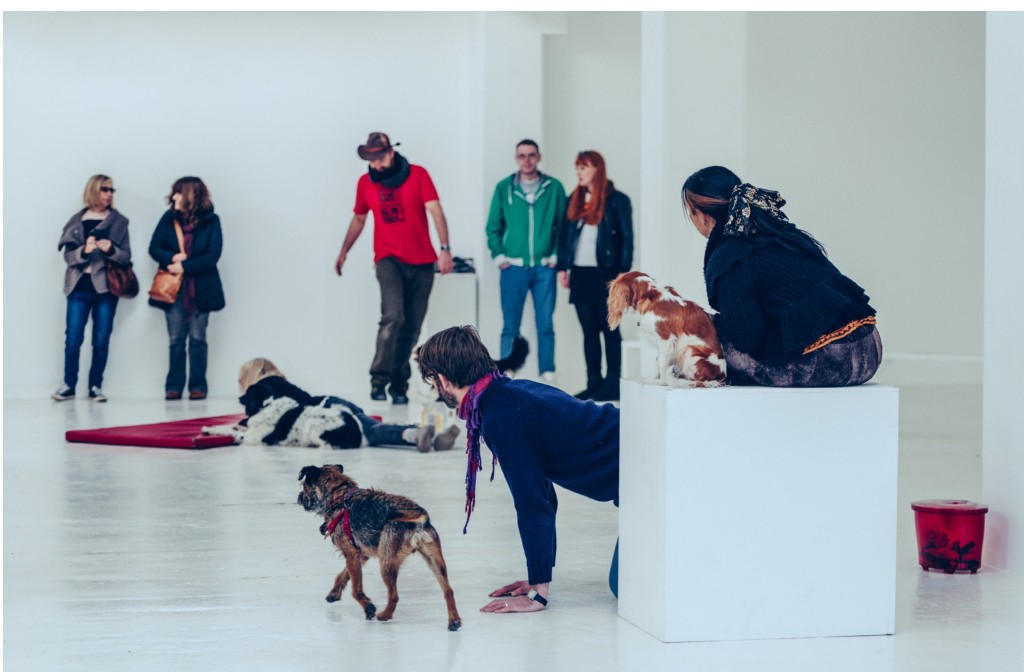

**Figure 4.** *Be Your Dog*, public performance, Karst Gallery, 2016. Photo by the artists.

Joseph Beuys' *I Like America and America Likes Me* was a three-day performance at Rene Block Gallery in New York in 1974 in which he and a coyote who had been, without consent, displaced from her habitat and life, resided together within the gallery. Beuys sought to engage and interact with the coyote in a human-centric and egotistical pursuit without consideration of benefit to the canine involved. It may appear to the viewer that towards the end of the event the coyote submits briefly to Beuys' touch. It was a spectacle intended to 'join' the human and the wild not-human in solipsistic service to embedded human political meaning. The coyote was a symbol, their presence meant to signify transgression even as the singular trespass was against their bodily and emotional autonomy. The coyote was the lesser member of the temporarily constructed pack of two.

The human participants' remit within *Be Your Dog* was to 'be' their dog. They were to observe: if the dog runs, the human should run; if the dog lays, they should lie down alongside them. This was not 'becoming animal' in the sense of Deleuze and Guattari, it was not a shifting from one to the other, but an acknowledgment of both species as contiguous and equal in contribution and performative agency in spite of, and through, their recognisable differences. Dogs are extraordinary empaths able to read emotions and feelings from others through their heightened senses, and it is this capability and connection that informs their abilities as companions. Asking the human participants to move and act with their dogs reasserted the connections between them as companions, as well as in the wider pack. They were asked to follow and adopt the positions their dogs adopted, and to observe and mirror what they do with their bodies, following their interests, and lines of vision. The dogs were the leaders—they were the defining artists within the constructed pack.

The animal behaviourists Elisabetta Palagi, Velia Nicotra, and Giada Cordoni conducted experiments to test if hierarchy in human-dog companion relationships is effective. Their studies, as described in their essay *Rapid Mimicry and Emotional Contagion in Domestic Dogs*, suggest that equality, based on a recognition of interspecies empathy within cohabitation, is more productive. Specifically, they link companionable canine attachment to humans with empathy, stating, "emotional contagion, a basic building block of empathy, occurs when a subject shares the same affective state of another" (Palagi et al. 2015, p. 2). *Be Your Dog* sought to explore and extend this suggestion, by locating the theory within the realms of artistic agency and collaboration, and by analysing how reflexive and assertive empathetic engagement within interspecies pairs can inform pack dynamics within the context of creativity as exercised within a gallery. The intention utilises this basis to test, and advance such scientific conclusions through situated artistic research, by seeing if and how the dogs would and could become artists both in collaboration and through recognition by others as such. How might the dogs respond to the site, and their associated positioning within its structure, as artists and creative makers? How would others view their inclusion, and would they acknowledge their contribution and artistic agency from the conditions of the site and performance itself? Through engaging in mutuality and co-empathy, will the creative position of all (dogs, humans, the gallery, the public performance, its viewers) become evident?

In shifting the balance of creative agency from the (normally problematic) anthropocentric within the gallery to an interspecies coalition, a heightened and effective dynamic was given to the creative act itself. By the end of the first day, with every interspecies body in their exhausted state, this new group started to function as a pack. Dogs were lying and mixing with different people than their own; there was evident interplay, without hierarchy being consuming and present. The mutuality and reciprocity of equal position was enhanced in vitality in the public performance, which saw the dogs and the whole interspecies pack respond positively to the audience. The try-outs during the workshops would last for approximately thirty minutes before the dogs decided to stop (as there was no coercion in play), by either lying down or leaving the room. Excited by their extended pack in the public performance, the dogs initiated and enthusiastically engaged for an hour and forty five minutes. Here, the dogs demonstrated their role as performers and collaborators, as artists who work to, and for, the observations of an audience. Importantly, all this took to produce was a sharing, a reciprocity of empathy, and the right conditions, site, and circumstance for their inclusion to be of acknowledged value. The installation in the exhibition of video footage of the public performance is purposefully and precisely set on the floor as dog accessible and dog referential, the 'body' of the monitor, with its electric cable 'tail' reflecting the mass of a large canine at rest (Figure 5).

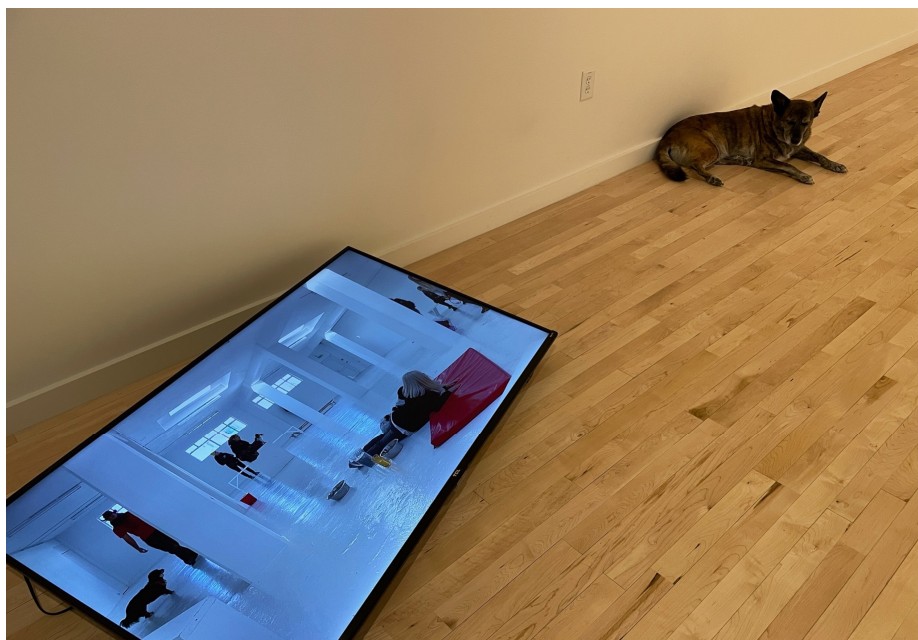

**Figure 5.** *Be Your Dog*, floor mounted monitor, with Elvira in the *Draw | Breath | Animal* exhibition, Tippetts and Eccles Galleries, 2021. Photo by the artists.

If dogs are unusual visitors within the confines of cultural walls, then more can be said of that in regard to a horse. Dogs are at least proximate anthropocentrically conditioned beings, who co-share human domestic spaces, but due to their size, scale, and outdoor living requirements, this is not so with a horse. To invite a live horse into a gallery or museum is an unusual spectacle, one which artists such as Janis Kounellis in *Twelve Live Horses* (Galleria L'Attico, Rome, 1969) and more recently Marion Laval-Jeantet, who received a transfusion of equine plasma direct from a living donor whilst mimicking its physicality by the wearing of hoof shoes in *May the Horse Live in Me* (Kapelica, Ljubljana 2011), have exploited. Similar to the non-coercive and uncontrolling propositions explored by Bartram in *Be Your Dog*, Deigaard creates a comfortable and welcome invite for animals to be within the gallery or museum. Like Bartram, and also in opposition to Beuys' interactions with the coyote as previously explored, Deigaard invited horses into a museum without coercion. Beuys shipped the coyote in against its will, and held a cane throughout his performance as an exercise in showing his control and dominance. Both Bartram and Deigaard use no such devices, and simply offer an invite (which is open to being ignored).

*Gus and Deuce Go Elsewhere* was made through a residency by Deigaard in 2014 at Elsewhere Museum in Greensboro, North Carolina, which invited artists to interact with their permanent collection, an immense amount of plastic, clothing, fabric, toys, and other wonders. Self-described as a living museum, the museum is housed in a former thrift store replete with all its original wares, and its multi-story building is a rich palimpsest of artist installations interacting with densities of object collections among which the artists also lived. As museum policy, object reverence, regardless of provenance or rarity, was democratically applied. If something accidentally was broken its pieces were preserved as part of the permanent collection. The museum presented a complex environment in which living and working required mindfulness against creating inadvertent chaos, and through inviting them inside this created an opportunity to explore the proprioception, sensory, and imaginative worlds of horses. Would they choose to come inside, and what will they make of this space? Deigaard's collaborative work often explores aesthetic and sensory experiences for interspecies animals to share, aiming to deepen mutual understanding and intimate connection. She went out into the field (literally) to meet and 'interview' horses in order to identify who might appreciate this kind of novel experience (knowing that some horses are curious about what is behind closed doors and others are not), and

ultimately invited Gus and Deuce and their humans to visit the museum. The project for Deigaard was not about getting horses inside a museum as a human spectacle (although it is acknowledged that, as with Bartram's artwork, the unusual nature of the invite to the non-human to step over the threshold into the interior of a cultural space can be seen as an occurrence that is spectacular), but to provide potential spectacle for the horses, in a horse-guided and horse-initiated art experience that also prioritised their emotional and physical well-being. The project began simply as an invitation for the horses to look through an open door and decide whether to come inside. If they chose to enter, they were free to poke around at will, their interests and curiosity guiding their exploration without cues or expectations or predetermined outcome. In the event, the horses stayed for more than two hours: touching and smelling, walking, hooves clopping, between rooms, standing at tables, examining art installations and novel objects from high shelves to the floor. They had entered freely as collaborators, as artists, and as viewers.

This project, like *Be Your Dog*, considers reverence (and its opposite): what a museum or gallery is, what is the art, who activates the art and how. The few people at Elsewhere Museum that day were there in primary service to the animal experience of art and artistic expression. As with Bartram's work, the horses' relationships with humans (as with each other) informed the project. A prevalent sense of interspecies awe and excitement pervaded the space. As the horses explored, they looked to one another for buttressing and support and sometimes in sibling-like competition over pieces of the permanent collection deemed desirable such as a particular plush toy. They looked also to their humans and to the artist whom they had met before in curiosity and shared experience. They interacted fully with the collection and art installations: smelling, touching, and even operating parts. As one example, Gus opened one antique cash register with his lips and later repeated the manoeuvre with a differently functioning register across the room (Figure 6). Both Deigaard and Bartram's work underscores empathetic awareness and imagination between species; curiosity practised in reciprocity and the full expression of animal personality, initiative, and preferences. It is based in improvisatory exploration and looks closely at play as a crucial component to research, imbuing it with revelatory potential. *Be Your Dog* and *Gus and Deuce Go Elsewhere* operate as an antithesis to the spectacularisation and coercion of Beuys' coyote, being constructed through invitation and without human control.

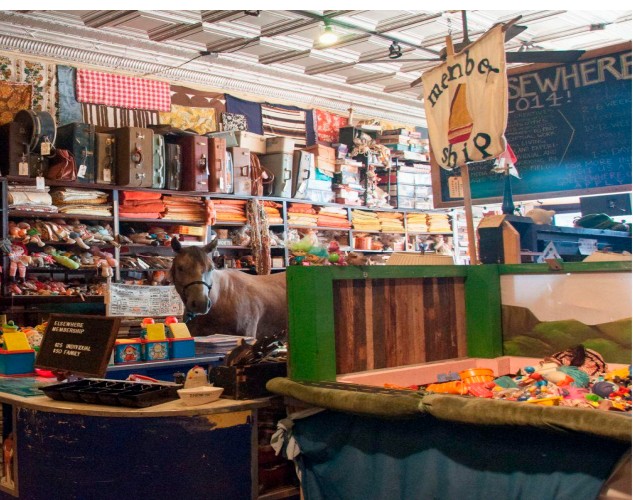 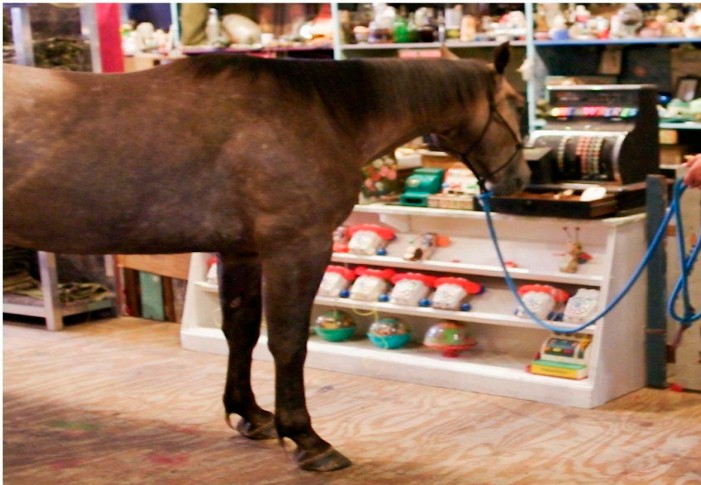

**Figure 6.** *Gus and Deuce Go Elsewhere*, supplementary project documentation, Elsewhere Museum, Greensboro, NC, 2014. Photo by the artists.

The context of the site is important in both artworks. Had either happened in a park or a field, the dogs and horses would not naturally be seen within art and potentially as artists. The gallery and museum are crucial; a visitor assumes, and potentially expects, that performing bodies and material things placed within their walls belong to artistic,

and intended, acts. Douglas Crimp reminds us in *On the Museum's Ruins*, that art appears "autonomous, alienated, something apart, referring only to its own internal history and dynamics" within a gallery or museum, but certainly as art (Crimp 1993, p. 13). The animals in these two artworks could, therefore, automatically be positioned as the artists because of the confines of their respective sites and its governing principles. However, the imposition is a trespass into a human-principled and directed institution of knowledge and creative exchange. Additionally, the presence of the non-human animal, logistically ephemeral and sensational in multiple senses, identify a tripartite way of being which is to say, as audience/participant, as artist, and as enactor/embodiment of art. Animals have aesthetic experiences: the dogs and horses were also art appreciators, investigating and performing within each site. By virtue of being animal, as artists Bartram and Deigaard aspire to be, they change and activate and heighten and elevate experiences of, and recognitions of, what art is and who artists are.

## 6. Contiguous Breathing

Breathing for mammals is a significant act. Not just in the obvious sense that it is necessary for survival, but when autonomous and sympathetic they are mostly unaware of it happening, unless it is laboured, difficult to stabilise, or becomes a focus for assured health. Put simply, mammalian bodies just get on with it, eliciting an unconscious adaptive response to the environment and stimuli. In ideal relations, a collaboration also breathes in unity, as an entity made of separate but conjoined parts and activity. Empathy and sympathy are acts of imagination, in part, and in this sense may be aspirational. At the same time, they arise organically and involuntarily due to proximity and accumulating awareness of respiration and pulse when other bodies lie near enough for these rhythms to be detected rather than imagined. Abiding with another, receptive and present, without predetermined outcome builds synchrony and enhances bonding with lifeforce and purpose. This is true also of artistic practice, which Bartram and Deigaard argue needs to metaphorically breathe in its creation and through its intended research inquiry and purpose. For the artist and the artwork, the gallery is an apparatus that acts as a continuation of breathing through an idea. The artwork discussed within the context of this text is certainly an affirmation, as it integrates and connects with this position within its infrastructure from idea to resolution.

Breathing involving connected and facilitating mammals and art practices informs the collaboration of Bartram + Deigaard (the '+' equating to becoming a collaboration), both in relation to being animal and through the exhibition of artefacts. From outset to implementation of this first collaborative exhibition, breathing and being reflective of, and engaged with, breath was instrumental in forging its direction. This was expanded into the gallery during installation, which saw long acts of spatial alignment and synchronous positioning in installation (including one seventeen hour day) to allow the artworks to settle into rhythm with each other (for example, the overlaying audio of their respective videos curated in proximity to each other, in which the percussion of horse hooves on wooden floors mingles with dog movements and human voices; these sounds combining with the noises of activity within the gallery itself, human and dog).

Bartram, alongside her continuing artistic research with dogs and mammals exploring equality in social and creative potential, has a contiguous interest in the methodologies of art-making as a being alive and perpetually in- and with-process that is of and in breath. This is often through the use and subversion of traditional means of art-making, such as etching and casting, whereby the process and material is given agency to respond to its conditions and evolve, by taking it beyond its disciplinary confines into a more transgressive (and thereby animal) state. Similarly, Deigaard often makes objects, drawings, and videos pitched to a non-human audience as vital and respondent. This audience can be single individuals with artworks made only for them; it can include reciprocal and layering processes whereby multiple and sequential versions of the works based on spontaneous animal interactions develop iteratively from subsequent interactions between animal and art.

An affirming reflection occurs, through their introduction (historical) and becoming and being a collaboration, by including their individual art videos, *Be Your Dog* and *Gus and Deuce Go Elsewhere*, and through the staging of new durational artworks that respond to the breath and breathing via given situations and within specific parameters. Each individual endeavour within the exhibition followed a durational pathway and commitment to the act of close observation and detailed working through of methodological constraints of medium and circumstances. They were solitary but connected pursuits of engagement with other presences, either via process and its potential mutabilities or through direct observational and/or experiential translations. Endurance and consistency (often critical not only in collaboration, but also in studio practice and the mounting of art exhibitions), are also part of the contract of love with the companion animal: being reliable, keeping to daily walks, and commitments of care. Trust grows in inter-reliance, togetherness, and attunement (just as with the artists). Between them are explored, consciously and unconsciously, concepts of shared breath and processes of inhalation and exhalation implying intimacy and proximity, but also their improvised ways for collaborating via a 'shared brain', like a respiratory limbic system. A breath that is mammalian and animal, and not only the preserve of the human.

The new artworks each artist contributed were large in volume and long-term durational projects originated and sustained through daily commitments. *366:366 (Finally)* by Bartram was a series of three hundred and sixty-six etchings made from the same plate, etched daily by her three hundred and sixty-six cumulative exhalations onto its surface (Figure 7). It constituted over four years of activity. Deigaard made over six hundred drawings, *Quarantine Drawings* (Figure 8) in ink on index cards of her sole companion in lockdown Elvira, between 17 March and 19 May 2020 when New Orleans was a pandemic epicentre. Both projects fit easily into small, portable containers: a shoebox of cards and a small portfolio of etchings carried onto a plane. A shared pursuit by individual means of close attention and detailing, of being animal (in the behaviours of releasing and recording breath), and of observing the animal (through figurative representation).

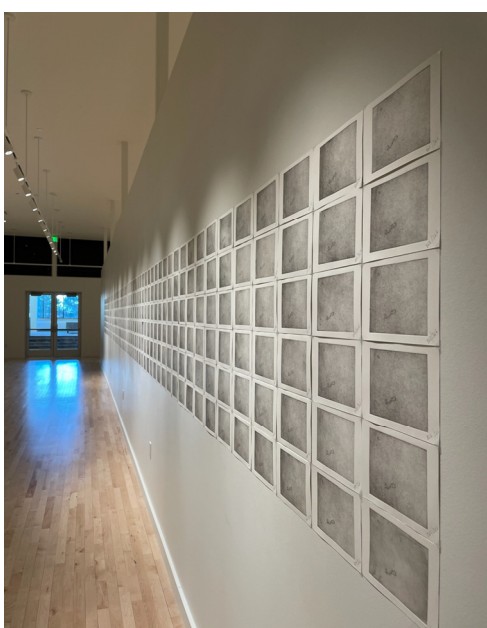

**Figure 7.** *366:366 (Finally)*, in the *Draw|Breath|Animal* exhibition, Tippetts and Eccles Galleries, 2021. Photo by the artists.

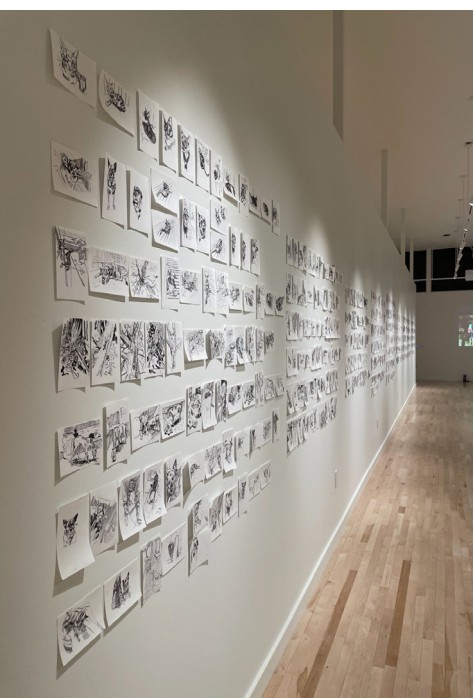

**Figure 8.** *Quarantine Drawings*, in the *Draw | Breath | Animal* exhibition, Tippetts and Eccles Galleries, 2021. Photo by the artists.

Bartram made her three hundred and sixty-six prints from a single etching plate. She exhaled onto its surface every day in 2016 which was a leap year and chosen for its rhythmic oddity and even number. The mouth symbolic and animal, a speechless and positioned maw for the artist in this context, behaving its mammalian and autonomous urge to breath. Against all the odds (experts told her it was impossible to etch with breath alone), her three hundred and sixty-six open-mouth deep daily exhalations upon the etching ground, through amalgamation and codified repetition, in fact engraved a clear image onto the A5 sized plate. Animal persistence bore result. Over the next three years, between 2017 and 2020, she made three hundred and sixty-six prints from that plate, an act, in its inclusive entirety, that constituted a reclamation and reactivation (an inhalation) of these breaths through the plate's erosion as well as deposition upon the paper in ink and stasis. With every run through the etching press, the plate degraded, losing depth (what holds the ink darkest) and the accumulating images also lost depth (and darkness), becoming ghosts of the initial corresponding actions and sequence now long past. The breath, through repetition and process, was eventually lost.

Inspired by Lynda Barry's five-minute drawings and her workshop *Drawing and Writing the Unthinkable*, Deigaard's daily drawing routine during the initial nine-week pandemic lockdown reflected on an interspecies relationship moving through the city, skirting peripheries, and evading human proximity. It documents a close observational and corporeal reliance on bodies permitted to touch during this time. Breath and voice, communicative and intimate, were illuminated and underlined as transmissible aerosols. The series turned around this moment, in a way, this physical manifestation of the breathing we must not share or intercept and the distances between bodies. Consequently, they speak to an intimacy, where the human is intently detailing their co-animal life in a reality distanced from other humans. Deigaard aimed to complete twelve ink drawings daily. The series began as a being within a companionship of parity and mutual animality in actions of primary and reciprocal caregiving: cohabitating, walks, what each reveal and shows to the other, how their curiosities informed and overlaid one another's. The drawings were sustained observation (and a kind of memorisation through drawing) of their movements and actions, outer and inner. Elvira was wholly present (and determinant of their unfolding)

within the activities depicted; she is there within the drawings themselves as she was there during their making in continuance and companionate support. Deigaard thought of the drawings as a practice (and demonstration) of love and devotion towards her dog: to study her and ponder her quicksilver reflexes, curiosity, and intelligence. The drawings consider Elvira as a singularly important observer and protagonist and connect them both, in the one-to-one, to shared breath. Finally, she was in the gallery for their installation and exhibition.

Each project comprises daily regularity and anchoring and a means of momentum required that culminates, through a reading of Elizabeth Grosz, as a 'body' engaged with the animal. Each body comprises and extends to the actual living bodies of its creative makers (whether human or non-human, in behaviour and observation): "the ideal in the material and the corporeal" subsists for Grosz (Grosz 2017, p. 5) and is maintained through the connection of matter to maker. Working with animals and unique personalities can be like working at the extreme parameters of a traditional means of making such as printmaking. Both have a semblance of unpredictability, and both require allowances and assurances for its inclusion. Failure, and its capacity to be prolific through the unpredictable and the un-prescribable, through change and duration, finds acceptance in such artworks. There is an added dimension to the stamina and devotional sacrifice for the process and subject required of the artist, as they must accept that the outcome and conclusion may have an end that is unknown.

## 7. Being Inside

The presence of the non-human animal is usually and normatively confined to representational artworks rather than the animal herself in the gallery or museum. Why is this so? If the question of the animal is so interesting a discussion to explore within a visual and critical context, then why is non-human representation only allowable if it is benign? Sure, observing animals in imagery is safe and informative, but it also negates the experience that is portrayed as being offered as a first-hand encounter. This spectacularisation of the non-human disallows any right- of and to- direct experience and connection, which is often a circumstance of gallery rules and legislation. A trespass of the other, the non-human, is as if a boundary has been breached to the danger of all others who may normally rely on being culturally safe in galleries and museums. Whilst these spaces are willing and open (and in some cases, inviting and needful) to curate artworks with non-human animals as subjects, as a welcome addition to public cultural offering, they often fear their physical presence within their walls. For there is an interest in the other, of observing difference, particularly that which is close to the human sense of being, but only at a safe distance and through representation, either moving or still image, material, object, or semblance alone. Direct experience, perhaps, is suggested as not being of interest to the artworld and its visiting public. Yet domestic companions are interested in human life enough to share its day-to-day environments and life. A barrier exists, however, to a dog or cat or horse receiving an invitation to enter. Fear of misbehaviour, or of inappropriateness (toileting, or being noisy, perhaps) or of others' imagined anxieties towards other species abides. Such thinking stalled Bartram's *Be Your Dog* happening for five years, for example. Galleries would simply not allow the dogs 'in'.

The invitation to a non-human to enter a gallery space is therefore incredibly unusual. If it happens, it is often not for the animals to explore and to enjoy culture, but for them to be a spectacle there within. Empathy and equality are significant to Bartram and Deigaard on all fronts when working with non-humans, as they wish to give them free access to cultural artefacts and happenings. To make this happen, each has negotiated a way in for their respective participants irrespective of their species as an important and significant move to real inclusion, both as viewer and as collaborator. Beyond the necessity of this within the making of the artworks *Gus and Deuce Go Elsewhere* (Deigaard) and *Be Your Dog* (Bartram), their collaboration seeks to give equitable and fitting access to the artworks and their content.

In *Draw|Breath|Animal*, the two videos were exhibited respectively at the eye level of horses as a wall projection (Deigaard's artwork) and on the floor where dogs (Bartram's artwork) could freely look. Deigaard's companion dog joined the collaboration in the gallery for this exhibition, aligning herself (along the upturned monitor) during their joint lecture as well as with the artists, and roamed freely within the space during installation of all the artworks included. In the gallery, their large-scale projects *366: 366 (Finally)* and *Quarantine Drawings* were installed simultaneously along a long, fixed wall, a central spine dividing the gallery through its centre. Each could not see the other, but they worked in rhythm to enmesh as collaborators, only as far apart as the depth of the wall and without speaking to the other. Installation, too, was a test of endurance and devotion; the dividing exhibition wall was fifty feet long, and each artist hung hundreds of sheets of images simultaneously over a continual seventeen-hour installation period. Throughout the installation, Elvira was activating the process; she was encountering the work in the space as audience, as participant, as subject matter, protagonist, and muse. (Figures 9 and 10).

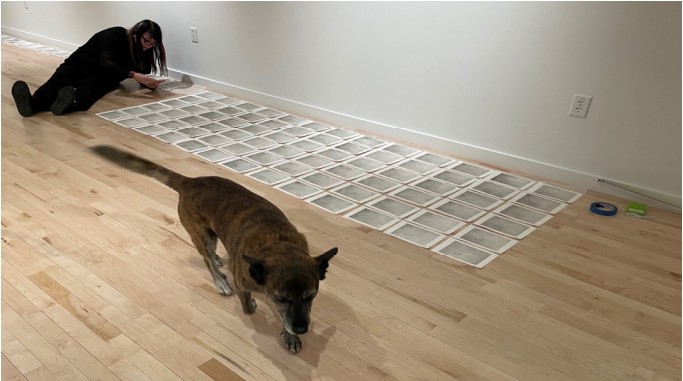

**Figure 9.** Installing *366:366 (Finally)*, with Elvira, in the *Draw|Breath|Animal* exhibition, Tippetts and Eccles Galleries, 2021. Photo by the artists.

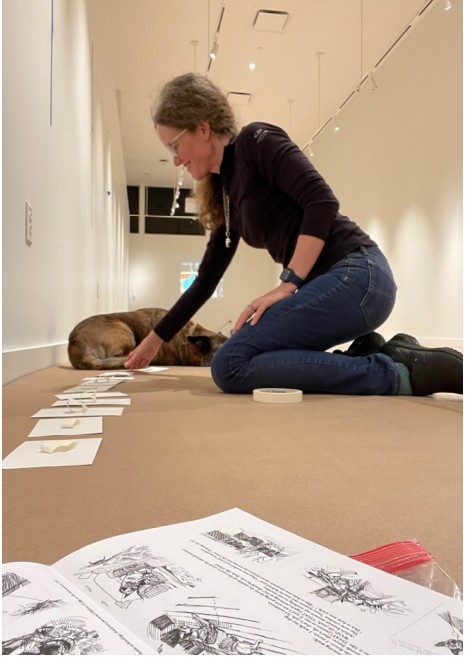

**Figure 10.** *Quarantine Drawings*, with Elvira, in the *Draw|Breath|Animal* exhibition, Tippetts and Eccles Galleries, 2021. Photo by the artists.

As the hours elapsed, Elvira was attuned to different emotional states and their mounting tiredness. They were asking endurance of her, too, and, for her part, she sorted what her role would be. This convergence along a central gallery floating wall or spine marked arduous and long individual journeys of cancelled flights and enforced layovers and of driving three days from Louisiana through thirteen states. Within the gallery Elvira moved at will, thereby (re)connecting the artists through her collaboratively binding presence. It is worth stating that, as in most requests to have dogs within a gallery, negotiations took place with a resulting contract being issued agreeing terms (days, times, events) of her presence which the artists countered, requesting if and whenever they were present, Elvira would be, too. During the lecture and through installation, there was a two-way gaze reciprocally between speaker/artefact and listener/observer (as there was with Elvira who stood naturally with the artists and with their work). She did not speak aloud as they did; only once did she use her voice when a masked visitor stared intently at her, but she underlined pointedly and deliberately with her body the rhythms of the work itself. Her presence heightened and delineated the exhibition of the work that had invited the animal into the museum or the contemporary art gallery. Elvira was a transmitter like 'commissural fibres' between brain hemispheres, but in this case, she held their shared brain with hers as a contiguous animal.

## 8. Three Female Animals Bonded by the Multiple Synchronies of Sharing and of Being Allowed In

The artists admit freely that installation of the exhibition would have been an entirely different experience without Elvira's presence. If distance is measured in miles and hours of travel, the artists narrowed an immense gap where they could connect through a conscious repetitive and adjacent working practice, of breathing between walls, and being connected through her visitations to each. They began their installation without prediction for how long it would take; they started at the same end and somehow ended organically in perfect synchronicity. Elvira connected them, at times watching them both from her bed. She was often with Deigaard and every now and then would check on Bartram, whom she had only just met. Bartram, while working in a fixed way along the wall, would suddenly feel a presence next to her that was very warm and gentle. Elvira vortexed around and for them both; they were connected, and she reminded them and underlined their connectivity.

Elvira was not welcomed into the gallery without discussion and permission being negotiated and obtained through the correct institutional channels of legislation. The institutionally necessary 'agreement' sought to bind the human artists into a conditional contract, but it was necessary for access to be granted. The artists, though, were bound further to their canine collaborator by the system and its rules—they became seen as more animal through this association. Even though this was perhaps not the gallery's intention, there was a dehumanising effect on the humans who were now aligned with their animal state. A lack of trust in potential behaviour and wariness as to their intentions was the implication through the enforcement of a contract. A performative re-animal-ing of the human artists was, in effect, instrumentalised by actions binding them to a contractual arrangement based around their shepherding and care (to not misbehave) of their canine. An interesting situation, whereby their associated difference as artists, following Foster's notion of the artist outsider-other, becomes heightened (Foster 1996, p. 173). Artistic commitment necessitates being outside of human sensibilities to make commentary and reflection upon existence for Foster, so perhaps this is inevitable. Yet the gallery, through its imposition of law regarding entry by the non-human, effected an animalising and othering of the humans they invited in as artists. In Foster's terms, it ensured they were seen as other.

Proprioceptive-ness, being mindful and responsive to sense and sensing, is keenly practiced by nearly all animals. It is necessary for survival; it is a beautifully tactful means of being thoughtfully and physically present with another. Those who are vulnerable employ it with additional burden. As a result, the state of being extremely responsive and aware can land disproportionately as an expectation and a responsibility on the less powerful.



Since human women, within patriarchal structures, still often experience an inequality and lessening in relation to men, it can therefore become intentionally gendered and part of the othering that so often takes place in the positioning and presence of women, their claiming of space and equal status. The artists examine structures of status within and between human and non-human animals. Three females together, irrespective of species, can create an unequivocal and equilateral affirmation forged of strength as a collective other. United in difference, dogs and humans in this context can collaborate through sharing and eliciting the effects of empathetic recognition and resolution to mutual and positive experience. Their difference becomes a focus, casting a spotlight on the collaborating components and their respective actions, divergences, and similarities.

So much of entering spaces entails this relationship between trespass and invitation. Is it explicit; is it implied? Who may enter? Many of us have a sense of trespass or intimidation when entering formal or 'high' art spaces, but this is necessary to some degree to gain the intended heightening in the experience of art including its making and presence. The experiencing of artworks is significant for Crimp. When regarding sculpture, he states that to reduce it "to the flat plane of the [reproduction] photograph, you're passing on only a residue" of the artist's intent for the idea (Crimp 1993, p. 167). For Crimp, the experience of the object, or art experience first-hand, is significant to understand and gain knowledge of the artwork's aim. Entrance (sometimes implicitly trespassing), therefore, is necessary to fulfil artistic promise, as it necessitates a reflective situation for insight into the position of the artwork and bodies within the exhibition. For the other, and a sense of being other, impacts upon all humans through a recognition of, and assertive working with and through, the question of the anti-anthropocentric and pro-animal. In this context, the latter was directional: to give equal status to the non-human as a creative and receptive being, they must be free to contribute, or not, and have the capacity to collaborate with the two female humans.

If we enter with trepidation, even more so does the dog. Bartram + Deigaard had permission for Elvira to be 'present,' and this allowed them to let her be herself in the gallery (leashed outside according to law but never in the gallery), to be unrestrained through not being continuously tethered. Yet there is also an invitation (artists try to imbue their work with it, they seek to connect through the work) because they say 'come see, come share with us,' which Elvira easily mirrored, embodied, and enacted. It relied on a subversion (albeit through signed agreement) of the cultural contract such spaces offer, however. This contract concerns the implicit agreement that (human) visitors will perform in a non-intrusive way within the gallery, so that it may perform its role of cultural offering. Kathy O'Dell, in *Contract with the Skin*, acknowledges that places are negotiated through human-centric agency and determination of use. O'Dell writes specifically of the living body in this context (without specifying species), as creating a "work of art that emphasises perspective" due to its physical presence and its being over the observation of it alone (O'Dell 1998, p. 32). Non-humans are not considered precisely because they cannot affirm agreement to behave as required. They cannot agree, even implicitly, to not touch artworks, for example, or observe at a distance (generally, museum standing distance is one metre from the artwork, so to be close enough, but beyond arm's reach), and this potential breach creates anxiety and apprehension. It seems trust to 'do the right thing' is predictable only in anthropocentric terms, which may account for part of the problem of anthropocentrism in cultural spaces. O'Dell regards a viewing contract as reliant on a metamorphosis, however, where the "subject is continuously transformed into the object and then back into the subject through and with his [or her] physical interactions with the environment" (O'Dell 1998, p. 8). To propose that this translation of subject through object is mutable, and metamorphic, is to align it with an otherness that is animal and different, and suggestive of a shift from a human position to one without anchor. Even though we know that humans can default on this position, in fact, they may not even be aware of the etiquette required, they are given the benefit of cultural doubt despite this in a way that a non-human is not. Both the dogs within the gallery and the horses within the

museum were respectful outwardly, while conducting it under their own terms. The dogs and horses' acute properties of proprioception as a matter of being, and their perspicacity and manners from negotiating complex life in a canine pack or equine herd, transferring into the cultural space.

## 9. Reflections on a Conclusion That Cannot Be Concluded

The critical effect of the activity for the artists and for others shows what inclusion can do. A showing of difference, and of doing things differently, enhances and contributes to the critical landscape and the development of pertinent and revolutionary questioning. The improvisatory and responsive aspect, the being open to and not presuming to know that we do, becomes significant. Where the non-human animal is given the means and agency to take creative initiative, there is much to be learnt. This situation in the artwork of Bartram + Deigaard, in some ways, is a co-optive and perhaps subconscious purposeful adoption of Foster's idea that an artist benefits from willingly positioning as outsider, and that of Deleuze and Guattari of being responsive to other animal worlds through a sympathy to becoming even if this is never truly manifested. Bartram and Deigaard, singularly and collectively in this collaboration and exhibition, examine this through an assertive acting out, hopeful that empathy between species, not as a becoming but in appreciation of difference as well, will be worthwhile.

A significant part of collaborations, in respect of the artists discussed in this text and across and with other species, relies on an unquantifiable but demonstrative infusion of energy that comes from curiosity, delight, mutual comfort, novelty, exploration, and improvisation. So much of what animals learn about the world is through watching and witnessing, being mindful and observant, which is often subconsciously engaged through interactions and initiatives across and beyond the boundaries of species. Bartram + Deigaard have chosen to put this front and centre in their enrichments and multi-animal explorations of culture. Empathetic connection, which allows the inquisitive to flourish and the behavioural to be enacted, offers a creative choice, of acceptance, interaction, dislocation, but never fear or coercion, to exist. They propose to create opportunities for animal initiative and autonomy, believing that it is better for all to be with others, to 'co' within existence. Rhizomic practice, of multiplicity and differentiated entrances and gateways into the artwork, is the companion to empathy in this context, allowing an enhancement of receptivity through a layered and non-hierarchical means of access.

The ability to have choices, to exercise autonomy, builds connectivity within brains, between individuals, and across species within their precise offerings which elicit full allowance and agency of participation. Humans place many curbs and restrictions on the non-human animal in their world and its culture, particularly in cities, and Bartram + Deigaard open a dialogue within the field of art and animal studies that strips these away. They do this by creating artistic research from a principled equal-capacity-for-all remit, by seeing all contributing animals as artists and all developmental and exhibition spaces as negotiated places for all species to be recognised as such. An assertive non-becoming of human to animal, nor animal to human is evident, in response to Deleuze and Guattari's suggestion that "becoming animal does not consist in playing animal or imitating an animal, it is clear that the human being does not "really" become an animal any more than the animal "really" becomes something else" (Deleuze and Guattari 1987, p. 238). They see affinity and positions of equality as contributing to being artists, because participating species are accepted and not co-opted into, or borrowed for creative purposes, but allowed to be who they are freely and within their own right and body. In this way, they operate more within Foster's concept (through his reading of *Time and the Other: How Anthropology Makes its Object* (1983) by Johannes Fabian) of the artist undergoing a projection that constitutes their position as 'outsider-other' and of being comfortable with that positionality and reference in their ethnographic testing of the boundaries of acceptable animal inclusivity in cultural practice and placement (Foster 1996, p. 177). The ethnography in their work, of experience and by the testing of situations, is undertaken by all species and,

thereby, in this context, all outsiders can be considered as both other and artist. Trespass in hierarchy, of species delineations, of the usual privileged spatial, creative, and systematic human-centric zones of exclusion, is a necessary negotiation. It allows inclusion of different, and non-human creative bodies and minds, to suggest how a different approach can be useful for all. Perhaps, in this way, these animal co-workers are untamed and considered feral, as allegiance and alliance with their non-human others is made apparent. This is perhaps heightened by gender consideration, and archaic perceptions of woman's feral behaviourist capacity.

There is much to be said for the virtues of co-learning and co-being; what Bartram + Deigaard offer here is a way to see and experience its cultural effect. Yet this is only the beginning and deeply embedded traditions are hard to change, so there can be no definitive conclusion . . . yet. However, a conscious and continually enquiring sensibility may see this way of thinking, of practice, take root to exist across material and bodily artistic creation. In her proposition of ontoethics, Grosz suggests that it "involves an ethics that addresses not just human life in its interhuman relations, but relations between the human and an entire world", (Grosz 2017, p. 2). Here we have discussed how empathetic and proprioceptive engagement extends to material and process through dedication and commitment, to sensibility and feeling of and with the inert, such as in Bartram's *366:366 (Finally)* and Deigaard's *Quarantine Drawings*, to equal measure with the living and physical (Figures 11 and 12). Proprioception, then, sensing and being sensitive to the other as a non-other and as equal (irrespective of species, materiality, and process), unifies purpose in the doing and a shared un-reliance on the outcome itself. Seeing to experiment and experience before result or expectation is important here, for equality and reciprocity in mutual curiosity and confidence enables effective collaborations.

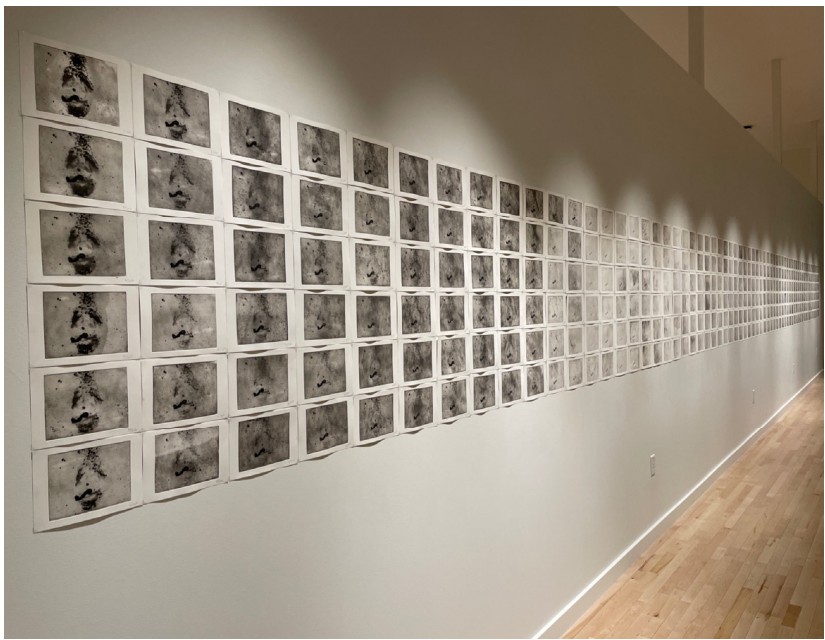

**Figure 11.** *366:366 (Finally)*, in the *Draw|Breath|Animal* exhibition, Tippetts and Eccles Galleries, 2021. Photo by the artists.

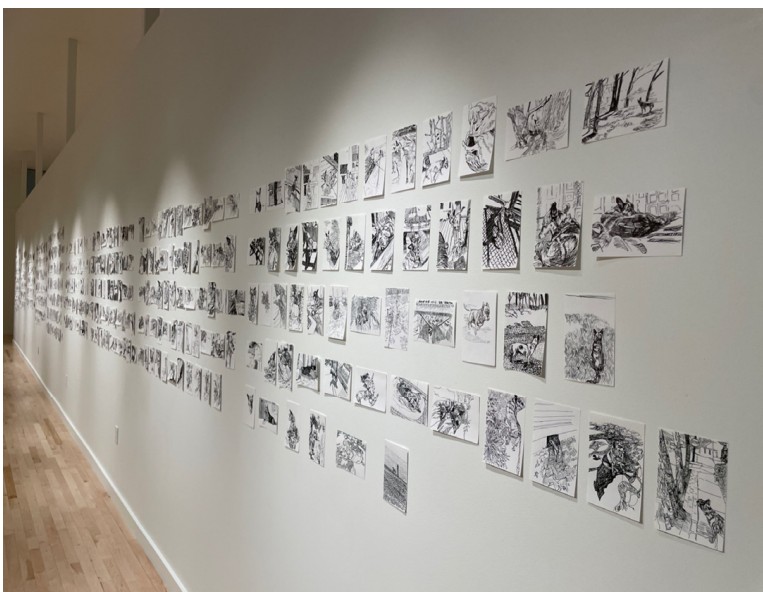

**Figure 12.** *Quarantine Drawings*, in the *Draw|Breath|Animal* exhibition, Tippetts and Eccles Galleries, 2021. Photo by the artists.

**Author Contributions:** Both authors contributed equally. All authors have read and agreed to the published version of the manuscript.

**Funding:** This research was funded by Utah State University, National Endowment for the Arts, and the Live Art Development Agency.

**Data Availability Statement:** Not applicable.

**Conflicts of Interest:** The authors declare no conflict of interest.

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
