# Peer review of "Shared Brains, Proprioceptiveness, and Critically Approaching the Animal as the Animal in Artworks"

_arts, 2023_

Round 1

Reviewer 1 Report

Your artistic contribution and ethical approach is very relevant and much needed in the art world and I really enjoyed reading your paper. 

There are just some minor things that could be improved: You should reference some of the scientific facts that you are stating (i.g. "animals have aesthetic experiences" or "to exercise builds connectivity within the brain"). Even if these things seem obvious to you, it adds some scholarly credibility if you can refer to a scientific study confirming your own findings or intuitions. For the same reason you might want to include some of the research on the bodily fundament for empathy (related to proprioception and maybe also mirror neurons) and the relation of art and empathy.

In line 298-299 referring to Beuys you are stating "Success was measured ... by the coyote submitting briefly to his touch...." The "touch" was a violent embrace and the coyote struggled to escape it. You should consider reformulate this sentence. And if there really is art historical literature that did measure success by this, you should refer to it.  Also, to my knowledge the coyote was male, at least his name was Little John. In line 301 and 302 you use female pronouns.

In line 365 you should state the title and venue of Kounellis "12 Horses" just as you did with Art Orienté Objet (actually Marion did not transfuse horse plasma herself, but Benoit Magin injected her horse plasma).

In line 552 respectively 689, you should mention Elizabeth Grosz' and Douglas Crimp's first name (as you need with Hans Ulrich Obrist, Hal Foster and Kathy O'Dell when first citing them).

Author Response

Thank you so much for the useful review. We have attended to the revisions, adding additional scientific references from animal behaviourists, added suggested extra information and reformulated advised sentences. 

Thanks also for saying you really enjoyed the paper, and for stating the work's relevance. 

Reviewer 2 Report

This is a strong, original article on an important topic and on artists whose work is important.

A few improvements might still reinforce the quality of the article. 

- Sometimes, especially in the introductory part, there are generalizing  phrasings as on  page 2 : "What animals can contribute with their minds beyond being material and ‘stuff’ is generally ignored ».

 Maybe some nuances might be added. The reading of books or articles showing non-human animals whose contribution « with their minds » is not ignored, could be useful (see below).

We can think about non human animals who have medical functions and help autistic people or others, often doing it without being asked by humans. 

 - p. 18). « Where the non-human animal is given the means and agency to take creative initiative there is much to be learnt ». The question of creative initiative in nature could be raised. The context of a gallery or a museum emphasizes this fact, but it could open on  a questioning about creativity without any human interactions (one can think about some Australian bird whose artistic talents, to ornate the front of the nest for the female, are evoked in some documentaries).

-Some words might be defined: concerning the use of the word “Pets” rather than “companions” at the beginning,  a brief comment on both terms would be interesting. And also a definition of Proprioceptiveness and Proprioception would be useful (in a note or in the text).

- Here are a few readings or documents that could be useful: 

Jacques Derrida, The Animal That Therefore I Am. Fordham University Press, 2008 [L’animal que donc je suis, Galilée, 2006].

Dona Harasway, When Species Meet, Minneapolis: University of Minnesota Press, 2007.

Dona Harasway, The Companion Species Manifesto: Dogs, People, and Significant Otherness, Chicago: Prickly Paradigm Press, 2003.

Scott Slovic, Zelia Bora, Marianne Marroum et al. Reading Cats and Dogs: Companion Animals in World Literature, Lexington Books, 2020.

Scott Slovic,  "Travels With Hanna: Dogs and/as Teachers”, 

https://www.canal-u.tv/chaines/universite-toulouse-jean-jaures/l-amour-des-animaux-animal-love/travels-with-hanna-dogs

Scott Slovic, "Dogs as Sensory Extensions of Self: A Gift”, 

https://www.canal-u.tv/chaines/universite-toulouse-jean-jaures/companion-species-in-north-american-cultural-productions-0

Corrections

p.3 , l. 110-111: "Any use of the terms henceforth, are made” —> is made

p. 7, l. 282 : "to not be"—> not to be

Author Response

Thank you so much for your thoughtful review, and for the comments to make it stronger. We have attended to the revisions, clarifying and bolstering points, as suggested, and adding in references form Haraway and Derrida (we originally had included these, but removed for word count reasons). 

Thank you also for commenting on the article's strength and originality on an important topic and on artists whose work is important.

Reviewer 3 Report

This is an interesting and generally well-written article about a topic that many animal studies scholars will find thought-provoking. I’m glad to see that someone is writing about Lee Deigaard’s work (I wasn’t familiar with Angela Bartram before reading this essay). For me, the strongest points of the essay are when you’re contrasting Deigaard and Bartram with Joseph Beuys, and when you’re discussing Gus and Deuce Go Elsewhere. In addition to making an original contribution to animal studies scholarship, I think this essay could spark some ideas for new artistic projects and stir up productive debate among artists who work with nonhuman animals about the ethical dimensions of their projects.

In my opinion, the essay would benefit from a rewrite carried out with the following points in mind:

It would make sense to start with a paragraph outlining the problem of the anthropocentrism of art galleries and museums—how, despite the pleasure that so many artists have taken in blurring human-nonhuman lines in ways that often produce discomfort in their viewers, they end up retreating to very comfortable human-centered positions in terms of keeping live critters out of their exhibition spaces. Some simple points would be welcome in this new opening paragraph about how museums and art galleries too often function as temples of anthropocentrism by enculturating their visitors to think of themselves as less animal-like than people who don’t appreciate art, and by presenting themselves as refuges from the living world rather than places to fully engage with it. Of course, along those lines, you could also add a few words about the long, long history of displaying taxidermy animals—how animals that are safely dead have always been acceptable and even quite desirable objects of scrutiny, but bringing a live nonhuman being into a museum or gallery would traditionally be considered scandalous.

* At various points when you’re talking about Elvira and Gus and Deuce and other animals exploring museums and galleries, I think it would be a good idea to further explain why these explorations are about more than just allowing animals to enter places that have previously been off-limits to them. I can imagine skeptical readers thinking, “So the artist brought her dog into the gallery with her. Big deal.” In what sense do these explorations constitute art? Maybe these parts of the essay could be enhanced by saying more about the impact these projects had on the animals themselves and on viewers. The photograph and your reflections on Be Your Dog on page 8 work well in this regard; I would like to hear more, for instance, about how the presence of animals in these projects led to greater relaxation (or more stress!) among the human participants.

* The transitions between your discussions of different projects by Deigaard and Bartram could be clearer. At some points, I had to do some work to ascertain which exhibition you were talking about. Maybe it would be helpful to say more about how ideas and problems associated with earlier projects informed the Utah exhibition—i.e., focus more on the narrative of Deigaard and Bartram’s development as artists. Also, straightforward overviews of each exhibition—exactly what it entailed—would be welcome at the beginning of your discussion of each one. That is especially true of what you write about the Utah exhibition.

Here are some comments on local issues in the essay:

Page 1: I would like to see the beginning of the abstract revised to mention the problem of anthropocentrism in galleries and museums, and it would be good to add references to some of Deigaard and Bartram’s other projects, since you spend so much time talking about them in the essay.

Page 4: The idea of the artists “becoming animal” definitely fits in here; I would just like to see more evidence and analysis scattered throughout the essay about how they have done this—and how they have managed to get viewers to rethink their own nature as animals.

Page 7: What you say about devotional practices is really interesting.

Page 8: The contrast between I Like America and America Likes Me and Be Your Dog is terrific. Has Deigaard said anything about Beuys? If so, that would be worth mentioning.

Page 9, lines 332-33: What you mean by “work itself out” isn’t clear to me. However, I like what you say at the beginning of the next paragraph about the formation of an “interspecies coalition.”

Page 10: I would like to see a contrast between May the Horse Live in Me and Gus and Deuce Go Elsewhere similar to the contrast on page 8. In addition to strengthening your claims, this would smooth out the currently bumpy transition between the paragraphs.

Page 11, line 446: I think it would improve the line “Breathing for mammals is a significant act” by adding something like, “not just in the obvious sense that it is necessary for survival, but…”

Page 12, lines 468-69: I’m not sure what you mean by “settle into rhythm in the interfering.”

Pages 12-13: I could use a simpler overview of what 366:366 (Finally) has to do with becoming animal and welcoming non-human beings into traditionally anthropocentric spaces, and how it complements/ complicates Deigaard’s Quarantine Drawings.

Page 14: “The presence of the non-human animal is usually and normatively confined to representational artworks rather than the animal herself in the gallery or museum.” Excellent—this is exactly what I think you should start saying at the very beginning of the essay.

In general, you could improve the essay by bringing some of the scholarly jargon down to Earth, although most animal studies scholars won't have any trouble with it. The main area for improvement, in my view, concerns your use of nominalizations and the passive voice when it would be better to begin your sentences with easy-to-visualize human or nonhuman subjects and build the sentences around strong verbs. The second paragraph on page 16, for example, features several sentences of the kind I teach my students to avoid. (I would be happy to add more comments on specific points of the essay if you would like them.)

Author Response

Thank you for your careful and insightful review of our article. We have attended to the revisions suggested, including opening the work with the issue of anthropocentrism in the gallery system, expanding on the unusual nature of animals being seen to be inside cultural venues, and rephrasing any unnecessary jargon. We have added more references to support parts of the work too. We have also attended to the precise revisions suggested on each page. 

Thank you also for saying this is an interesting, original and well-written article about a topic that many animal studies scholars and artists will find thought-provoking.